# Synergistic Effects of the Combinational Use of Escitalopram Oxalate and 5-Fluorouracil on the Inhibition of Gastric Cancer SNU-1 Cells

**DOI:** 10.3390/ijms232416179

**Published:** 2022-12-19

**Authors:** Vincent Chin-Hung Chen, Jing-Yu Huang, Bor-Show Tzang, Tsai-Ching Hsu, Roger S. McIntyre

**Affiliations:** 1Department of Psychiatry, School of Medicine, Chang Gung University, Taoyuan 33302, Taiwan; 2Department of Psychiatry, Chang Gung Medical Foundation, Chiayi Chang Gung Memorial Hospital, Chiayi 61303, Taiwan; 3Institute of Medicine, Chung Shan Medical University, Taichung 40201, Taiwan; 4Immunology Center, Chung Shan Medical University, Taichung 40201, Taiwan; 5Department of Clinical Laboratory, Chung Shan Medical University Hospital, Taichung 40201, Taiwan; 6Department of Biochemistry, School of Medicine, Chung Shan Medical University, Taichung 40201, Taiwan; 7Mood Disorders Psychopharmacology Unit, University Health Network, University of Toronto, Toronto, ON M5T2S8, Canada; 8Department of Psychiatry, University of Toronto, Toronto, ON M5T1R8, Canada

**Keywords:** hepatocellular carcinoma (HCC), selective serotonin reuptake inhibitors (SSRIs), escitalopram, autophagy, nationwide population-based cohort study

## Abstract

Owing to its high recurrence rate, gastric cancer (GC) is the leading cause of tumor-related deaths worldwide. Besides surgical treatment, chemotherapy is the most commonly used treatment against GC. However, the adverse events associated with chemotherapy use limit its effectiveness in GC treatment. In this study, we investigated the effects of using combinations of low-dose 5-fluorouracil (5-FU; 0.001 and 0.01 mM) with different concentrations of escitalopram oxalate (0.01, 0.02, 0.06, and 0.2 mM) to evaluate whether the assessed combination would have synergistic effects on SNU-1 cell survival. 5-FU (0.01 mM) + escitalopram oxalate (0.02 mM) and 5-FU (0.01 mM) + escitalopram oxalate (0.06 mM) administered over 24 h showed synergistic effects on the inhibition of SNU-1 cell proliferation. Moreover, 5-FU (0.001 mM) + escitalopram oxalate (0.02 or 0.06 mM) and 5-FU (0.01 mM) + escitalopram oxalate (0.02, 0.06, or 0.2 mM) administered over 48 h showed synergistic effects on the inhibition of SNU-1 cell proliferation. Compared with controls, SNU-1 cells treated with 5-FU (0.01 mM) + escitalopram oxalate (0.02 mM) exhibited significantly increased levels of annexin V staining, reactive oxygen species, cleaved poly (ADP-ribose) polymerase, and caspase-3 proteins. Furthermore, 5-FU (12 mg/kg) + escitalopram oxalate (12.5 mg/kg) significantly attenuated xenograft SNU-1 cell proliferation in nude mice. Our study is the first to report the synergistic effects of the combinational use of low-dose 5-FU and escitalopram oxalate on inhibiting SNU-1 cell proliferation. These findings may be indicative of an alternative option for GC treatment.

## 1. Introduction

Gastric cancer (GC) is the fifth most common cancer and the third leading cause of cancer-related deaths worldwide [1]. GC is commonly diagnosed at advanced stages because it is asymptomatic in the early stages and because reliable biomarkers for its identification have yet to be developed [2]. Despite recent advances in chemotherapy and surgical techniques, GC remains a life-threatening malignancy owing to its high recurrence rate [3,4]. The postoperative recurrence rate of GC is high even after radical resection, with a median survival time of 12.5–13.0 months and a 2-year survival rate of 22.9–23.6% [5]. 5-Fluorouracil (5-FU) is a common first-line drug against advanced GC [6]. Clinical data have demonstrated that 5-FU administration can increase the survival rate of patients with GC by 6% and reduce the risk of mortality by 18% [7]. However, approximately 21% of all patients with GC develop resistance to 5-FU, considerably limiting the clinical use of 5-FU against GC [8].

Mental disorders such as depression, bipolar disorder, schizophrenia, and dementia create a major global burden and have considerable negative impacts on health, society, and the economy [9]. Antidepressants and antipsychotics are commonly used to treat mental disorders [10,11]. Antidepressants can be divided into five main classes: selective serotonin reuptake inhibitors; serotonin-noradrenaline reuptake inhibitors (SSRIs); noradrenaline and specific serotonergic antidepressants; tricyclic antidepressants; and serotonin antagonists, reuptake inhibitors, and monoamine oxidase inhibitors [12]. Of these classes, SSRIs-fluoxetine, paroxetine, sertraline, and escitalopram oxalate are the most commonly used antidepressants [12]. Moreover, sulpiride, risperidone, olanzapine, and quetiapine are commonly used new-generation antipsychotics [13]. The association between the use of antidepressants and antipsychotics and the risk of cancer has attracted increasing attention recently [14,15,16,17,18,19,20]. Notably, a recent study reported that the use of antipsychotics such as thioridazine, clozapine, haloperidol, sulpiride, olanzapin, quetiapine, amisulpride, and risperidone is independently and inversely associated with the risk of GC [17].

Several studies have reported that certain antidepressants and antipsychotics exhibit anticancer activities. Imipramine and amitriptyline have been reported as adjuvant therapy for glioblastoma multiforme by switching the glioma stem cells (GSCs) to non-GSC [21]. A study of both bench work and a nationwide population-based cohort study also demonstrated the beneficial of escitalopram oxalate on hepatocellular carcinoma (HCC) [22]. However, a single antidepressant has limited efficacy against cancer. Therefore, combination therapy with two or more therapeutic agents has been adopted and investigated in cancer research [23]. Notably, a recent study indicated that sertraline and fluoxetine can synergize with sorafenib and suppress the HCC cells both in vitro and in vivo through blocking the AKT/mTOR pathway [24], indicating a synergistic effect of the combinational use of antidepressants and chemotherapeutic drugs. In the present study, we assessed whether the combination of escitalopram oxalate (Forest Laboratories, Inc., St. Louis, MO, USA) and 5-FU had synergistic effects on inhibiting the survival of SNU-1 human GC cells.

## 2. Results

### 2.1. Effects of Escitalopram Oxalate and 5-FU on SNU-1 Cell Viability

To verify the effects of commonly used antipsychotics and antidepressants on GC cells, we assessed the viability of SNU-1 cells in the presence of varying doses of quetiapine, clozapine, risperidone, and escitalopram oxalate for 24 and 48 h (Figure 1A). Escitalopram oxalate considerably attenuated the proliferation of SNU-1 cells (Figure 1); thus, the subsequent experiments were conducted using escitalopram oxalate alone or in combination with chemotherapy (5-FU). Escitalopram oxalate and 5-FU significantly reduced the survival of the SNU-1 cells in a dose- and time-dependent manner with the IC_50_ values of 0.27 and 1.16 mM at 24 h and 0.19 and 0.03 mM at 48 h, respectively. (Figure 1B).

### 2.2. Synergistic Effects of Escitalopram Oxalate + 5-FU on SNU-1 Cell Viability

We used low-dose 5-FU (0.001 and 0.01 mM) in combination with varying concentrations of escitalopram oxalate to assess the synergistic effects of this combination on SNU-1 cell proliferation (Figure 1C). The viability of SNU-1 cells treated with varying concentrations of escitalopram oxalate (0.02, 0.06, and 0.2 mM) + 5-FU (0.001 mM) for both 24 and 48 h did not differ significantly from those treated with escitalopram oxalate alone (Figure 1C). Notably, the viability of SNU-1 cells treated with varying concentrations of escitalopram oxalate (0.02, 0.06, and 0.2 mM) + 5-FU (0.01 mM) for 24 and 48 h decreased significantly compared with those treated with escitalopram oxalate alone (Figure 1C). Moreover, we calculated combination index (CI) values by using CompuSyn (CompuSyn Inc., Paramus, NJ, USA) to confirm the pharmacological interactions between escitalopram oxalate and 5-FU (Figure 2). The administration of 5-FU (0.01 mM) + escitalopram oxalate (0.02 and 0.06 mM) for 24 h exhibited synergistic effects (CI < 1) on the inhibition of SNU-1 cell viability (Figure 2A). Moreover, the administration of 5-FU (0.001 mM) + escitalopram oxalate (0.02 mM and 0.06 mM) for 48 h and 5-FU (0.01 mM) + escitalopram oxalate (0.02 mM, 0.06 mM, and 0.2 mM) for 48 h exhibited synergistic effects (CI < 1) on the inhibition of SNU-1 cell viability (Figure 2B).

### 2.3. Effects of Escitalopram Oxalate + 5-FU on SNU-1 Cell Apoptosis

Flow cytometry based on PI and annexin V/PI double staining analysis was performed to detect the cell cycle stages and the percentage of apoptosis induced by escitalopram oxalate and 5-FU. Compared with those in the control cells and cells treated with escitalopram oxalate or 5-FU alone, the sub-G1 portion in the SNU-1 cells increased significantly upon treatment with escitalopram oxalate (0.2 mM) + 5-FU (0.01 mM) (Figure 3A). The quantified results are illustrated in Figure 3B. To further confirm the presence of apoptosis, annexin V/PI staining and Immunoblotting were performed. Compared with the control cells and cells treated with escitalopram oxalate or 5-FU alone, the SNU–1 cells treated with escitalopram oxalate (0.2 mM) + 5–FU (0.01 mM) exhibited significant apoptosis (Figure 3C). The quantification results are presented in Figure 3D. Accordingly, a significantly elevated ROS level was detected in SNU-1 cells treated with escitalopram oxalate (0.2 mM) + 5-FU (0.01 mM) (Figure 3E). Significantly increased cleaved poly (ADP-ribose) polymerase (PARP) was observed in the SNU-1 cells treated with escitalopram oxalate (0.2 mM), 5-FU (0.01 mM) and escitalopram oxalate (0.2 mM) + 5-FU (0.01 mM), respectively (Figure 3F). The amount of cleaved caspase-3 was also significantly increased in the SNU-1 cells treated with escitalopram oxalate (0.2 mM) + 5-FU (0.01 mM). The quantification of Immunoblotting is shown in the right panel of Figure 3F.

### 2.4. Effects of Escitalopram Oxalate + 5-FU on the Growth of Xenograft SNU-1 Cell Tumors in Nude Mice

To further examine the effects of escitalopram oxalate + 5-FU in vivo, xenograft SNU-1 cell tumors were generated in BALB/c nude mice. Notably, the mice treated with escitalopram oxalate (12.5 mg/kg) + 5-FU (12.5 mg/kg) showed a significantly reduced xenograft tumor volume compared with the controls and the mice treated with escitalopram oxalate (12.5 mg/kg) or 5-FU (12.5 mg/kg) alone (Figure 4A,B). An apparent increase in the number of TUNEL-positive cells was observed in the xenograft tumor sections obtained from the mice treated with escitalopram oxalate (12.5 mg/kg) + 5-FU (12.5 mg/kg) as compared with those in the sections obtained from the controls and from the mice treated with escitalopram oxalate (12.5 mg/kg) or 5-FU (12.5 mg/kg) alone (Figure 4C).

## 3. Discussion

Although chemotherapy may diminish or attenuate cancer cell growth and reduce cancer recurrence, physical discomfort and drug toxicity remain the major issues of chemotherapy. A mounting body of evidence has indicated that the combinational use of multiple drugs is superior to monotherapy in the treatment of several cancers. This phenomenon is now known as synergistic interactions [21]. Compared with additive combinations, synergistic drug combinations afford more favorable therapeutic effects at lower drug doses [25]. In this study, we investigated the synergistic effects of 5-FU + escitalopram oxalate at lower doses on SNU-1 cell inhibition, the significant induction of apoptosis in SNU-1 cells, and the attenuation of xenograft SNU-1 tumors in nude mice.

5-FU is an antimetabolite that blocks the action of thymidylate synthase and incorporates its metabolites into RNA and DNA [26]. As the most frequently used chemo-therapeutic, 5-FU has long been utilized in regimens of neoadjuvant chemotherapy for various cancers [27]. However, several clinical trials have reported that regimens comprising 5-FU have a relatively high toxicity profile [28]. The most common toxic effects of capecitabine, an oral prodrug of 5-FU, include diarrhea, nausea, vomiting, stomatitis, and hand-foot syndrome [29]. DCF (docetaxel, cisplatin, and 5-fluorouracil), a chemotherapeutic regimen that comprises docetaxel, cisplatin, and 5-FU, causes adverse events, especially neutropenia, in patients with advanced GC [30,31]. Moreover, a recent study indicated that 5-FU upregulates exosomal PD-L1 levels and leads to systemic immunosuppression in patients with advanced GC following multiple chemotherapy cycles [32] In addition to these adverse effects, the rapid emergence of resistance to 5-FU-based chemotherapy is another major clinical problem associated with the use of 5-FU [33,34]. Therefore, to solve the abovementioned adverse effects of 5-FU, identifying methods for reducing the dose of 5-FU while increasing cancer cells’ sensitivity to 5-FU to effectively achieve anticancer effects is imperative. Notably, the present study indicated that the use of low-dose 5-FU + escitalopram oxalate exerts synergistic cytotoxic effects on GC SNU-1 cells by inducing apoptosis. Our findings provide a possible solution to the adverse effects of 5-FU in GC treatment.

Escitalopram oxalate is an SSRI and is known as an antidepressant [35]. Evidence indicates that escitalopram oxalate exhibits favorable tolerability and fewer toxic effects as compared with monoamine oxidase inhibitors and tricyclic antidepressants [36,37] Notably, cohort studies have reported that escitalopram oxalate is associated with reduced risk of various cancers, including bladder cancer, hepatocellular cancer, and kidney cancer [38,39,40] as well as the overall improvement in the survival of patients with GC [17]. Moreover, studies have reported that escitalopram oxalate inhibits non–small-cell lung cancer and brain tumor cells through apoptosis or autophagy [41,42]. These findings highlight the anticancer potential of escitalopram oxalate. Although a previous report indicated that the daily administration of 600 mg of escitalopram oxalate does not cause any adverse symptoms [42], some controversial cases of escitalopram oxalate overdose have been reported. For instance, a case report presented both QRS complex widening and QTc interval prolongation in a patient after an escitalopram oxalate overdose [43]. Therefore, a more favorable treatment approach is one that can achieve curative effects using a reduced dose of escitalopram oxalate. In the current study, the use of 0.01 or 0.001 mM 5-FU + low-dose escitalopram oxalate (0.02 or 0.06 mM) was found to have synergistic effects on SNU-1 cell inhibition, indicating the greater feasibility and superiority of escitalopram oxalate in GC treatment.

Very little information is known about the effects of escitalopram oxalate on drug-resistance. Interestingly, a recent study of bacterial drug-resistance reported that the combinational use of escitalopram oxalate with ciprofloxacin or sulfamethoxazole-trimethoprim had a significant synergistic effect on inhibiting multidrug-resistant (MDR) bacteria as compared with these antibiotics alone [44], indicating a potential of escitalopram oxalate on overcoming the drug-resistance issue. Since only one gastric cancer cell line, SNU-1, was assayed in this study, a 5-FU-resistant human gastric cancer cell line such as SNU-620-5FU [45] should be adopted to investigate the effects of combinational use of escitalopram oxalate and 5-FU on 5-FU-resistant human gastric cancer cells.

Although this study proved the combinational use of low-dose 5-FU and escitalopram oxalate on inhibiting SNU-1 cell proliferation, the underlying mechanism is still poorly understood. Notably, a study by using miRWalk analysis reported plasma miRs as potential markers for major depressive disorder (MDD) patients treated with Escitalopram, which could understand the Escitalopram mode of action and for its side effects [46]. Another in silico transcriptomic-wide association study was also performed to predict the transcriptomic profile of citalopram remitters [47]. Moreover, a very recent metabolomics study using LC-MS indicated indoleamine 2,3-dioxygenase 1 (IDO1) as a therapeutic target for pancreatic cancer-associated depression [48]. These findings suggest that genomic and proteomic approaches might have great potentials in understanding the mechanism and clinical benefit of combinational use of 5-FU and escitalopram oxalate on treatment of GC.

Some limitations of the present study should be mentioned. First, drug resistance is still a critical issue that limits the clinical utility of 5-FU [49]. Although the current study reported the synergistic effects of 5-FU + low-dose escitalopram oxalate on GC cell inhibition, the efficacy of such drug combinations on 5-FU-resistant GC cells remains unclear. Therefore, further research is warranted to assess whether 5-FU + escitalopram oxalate exhibits synergistic effects on GC cell inhibition. Second, pharmacological synergy is adequately defined in preclinical experiments, especially in cell line studies; however, relevant data are insufficient for conducting cancer clinical trials. Accordingly, further research is necessary to obtain a clearer understanding of the variability in drug response and novel biomarkers. This implies the need for preclinical research on diverse cancer models rather than focusing on drug synergy [23]. Third, the current study included only subcutaneous xenograft tumor experiments and thus could not identify the hallmarks of human gastric tumors, including tumor development, metastatic activity, and response to therapy. Therefore, as described elsewhere [50], an orthotopic GC animal model may need to be adopted for verifying the precise effects of the use of 5-FU + escitalopram oxalate.

## 4. Materials and Methods

### 4.1. Chemicals

The chemicals used in this study were of analytical grade and were obtained from Sigma–Aldrich (St. Louis, MO, USA). Escitalopram oxalate and 5-FU were provided by Chiayi Chang Gung Memorial Hospital, Taiwan.

### 4.2. Cell Viability Assay

SNU-1, the GC cell line, was purchased from the Bioresource Collection and Research Center and maintained in Roswell Park Memorial Institute Medium 1640 (Gibco, Brooklyn, NY, USA) containing 10% fetal bovine serum at 37 °C in 5% CO_2_. The 2,3-bis (2-methoxy 4-nitro-5-sulfophenyl)-2H-tetrazolium-5-carboxanlide inner (XTT) assay was used to determine the viability of SNU-1 cells. The cells were cultured in 96-well culture plates (5 × 10^3^/well) overnight at 37 °C and then subjected to varying concentrations of escitalopram oxalate and 5-FU administered alone or in combination for another 24 and 48 h. After the incubation process was completed, the medium was removed, and fresh culture medium was added. Subsequently, a total of 50 µl XTT was added to each well of the 96-well culture plates and then incubated for another 4 h (Biological industries, Haemek, Israel). Finally, the absorbance was detected at a wavelength of 630–690 nm by using a microplate reader (EnSpire Series Multilabel Plate Readers, PerkinElmer Inc., MA, USA).

### 4.3. Evaluation of Combination Index

The synergistic effects of the different combinations of escitalopram oxalate and 5-FU on SNU-1 cell survival were evaluated using the combination index (CI) proposed by Chou [51]. A CI value between 0 and 1 indicates synergism (more than additive effects). Moreover, a fraction affected (Fa) value of <0.5 indicates lower growth inhibition and is considered irrelevant. By contrast, a Fa value of >0.5 indicates a significant effect of the drug on the tested cancer cells.

### 4.4. Detection of Sub-G1 Portion

Flow cytometry was conducted to detect the sub-G1 portion of the SNU-1 cells. The cells were incubated with escitalopram oxalate (0.2 mM) or 5-FU (0.01 mM) alone or in combination for 48 h and then rinsed with phosphate-buffered saline (PBS), after which they were fixed in 70% alcohol for 12 h at 4 °C. Next, 10 μL of propidium iodide (PI) staining solution was added, and the mixtures were then incubated on ice under dark conditions. The cells were then filtered through a 40-μm nylon screen and subjected to flow cytometry using the FACSCanto II system (BD Biosciences; San Jose, CA, USA).

### 4.5. Detection of Apoptosis

The annexin V assay was used to assess cell apoptosis. A total of 1 × 10^6^ SNU-1 cells were incubated with escitalopram oxalate (0.2 mM) or 5-FU (0.01 mM) alone or in combination for 48 h. The cells were then centrifuged and re-suspended in 100 μL of annexin-binding solution. Next, 5 μL of annexin V–fluorescein isothiocyanate and 1 μL of PI solution were added. After being incubated under dark conditions at room temperature for 15 min, the stained cells were subjected to flow cytometry using the FACSCanto II system (BD Biosciences; San Jose, CA, USA).

### 4.6. Detection of ROS Level

Reactive oxygen species (ROS) levels were determined using the fluorogenic probe 2,7-dichlorofluorescin diacetate (Cayman Chemical, Ann Arbor, MI, USA). The SNU-1 cells were incubated in culture medium containing varying doses of 5-FU or escitalopram oxalate at 37 °C for 16 h. The cells were then treated with 50 µM 2,7-dichlorofluorescin diacetate and incubated at 37 °C for 30 min. Next, the fluorescence intensity at emission wavelengths of 502 and 523 nm was measured using a microplate reader (EnSpire Series Multilabel Plate Readers, PerkinElmer Inc, MA, USA) after the replacement of the incubation reagent with cell-based assay buffer.

### 4.7. Immunoblotting

Immunoblotting was performed as described elsewhere [22]. The presence of apoptosis-related proteins was detected with antibodies against PARP (Proteintech, Rosemont, IL, USA) and caspase-3 (Proteintech, Rosemont, IL, USA). Antibodies against β-actin (Sigma-Aldrich, St. Louis, MO, USA) were used as an internal control. Briefly, the cells with different treatment were harvested by centrifugation and lysed in lysis buffer (PRO-PREP™, iNtRON Biotechnology Inc., Gyeonggi-do, Korea). The protein lysates were transferred to PVDF membranes (1000 Alfred Nobel Dr Hercules, CA, USA) after separating onto a 12% SDS-PAGE gel. The membranes were then socked in 5% non-fat milk for 6 h with gentle agitation and subsequently reacted with antibodies at 4 °C overnight. After incubating with horseradish peroxidase (HRP)-conjugated antibodies, the presence of immune-complexes was measured with a chemiluminescent substrate kit (EMD Millipore, Burlington, MA, USA) and a chemiluminescence imaging device (GE ImageQuant TL 8.1; GE Healthcare Life Sciences, Pittsburgh, PA, USA). Additionally, a Multi-Gauge Software (Fujifilm Corporation, Tokyo, Japan) was used to quantify the blots.

### 4.8. Mouse Xenograft Model

Twenty male BALB/c athymic nude mice aged 5 weeks were provided by the National Center for Experimental Animals, Taiwan, and housed in a specific pathogen-free facility under constant environmental conditions with a 12-h light–dark cycle. All animal experiments were approved by the Institutional Animal Care and Use Committee of Chiayi Chang Gung Memorial Hospital, Taiwan (approval number: 2018080201). A total of 5 ×10^6^ SNU-1 cells/100 μL of PBS were then subcutaneously injected into the flank of the mice. The xenograft tumor volume was approximately 80 mm3. The mice were randomly divided into a control group (C), escitalopram oxalate group (L), 5-FU group (5-FU), and escitalopram oxalate + 5-FU group (L + 5-FU) and were administered PBS, escitalopram oxalate (12.5 mg/kg), 5-FU (12 mg/kg), and escitalopram oxalate (12.5 mg/kg) + 5-FU (12 mg/kg), respectively, by oral gavage on a daily basis. The tumor volumes were calculated weekly using a caliper and the mice were sacrificed after 6 weeks of treatment.

### 4.9. Terminal Deoxynucleotidyl Transferase (TdT) Mediated Nick end Labeling (TUNEL) Staining

Terminal deoxynucleotidyl transferase dUTP nick-end labeling (TUNEL) staining was performed using a commercial kit (Abcam, Cambridge, UK). The tissues were cut into 3 µm-thick sections and deparaffinized, rehydrated, quenched, and reacted with proteinase K. The TUNEL staining process was then performed using terminal deoxynucleotidyl transferase with digoxin-labeled dUTP in accordance with the manufacturer’s instructions.

### 4.10. Statistical Analysis

All statistical analyses were conducted using SAS JMP 7.0 software (JMP, Cary, NC, USA). We conducted a one-way analysis of variance followed by Tukey’s multiple-comparisons test to calculate statistical significance. All data are presented as mean ± standard error of the mean. *p* < 0.05 was considered to indicate statistical significance.

## 5. Conclusions

In summary, we reported that a regimen containing low-dose 5-FU and escitalopram oxalate showed a synergistic effect on inhibiting the proliferation of SNU-1 human gastric cancer cells, in particular the combination of 0.01 mM 5-FU and 0.2 mM escitalopram oxalate. As a result of the synergistic effects of combinational use of 0.01 mM 5-FU and 0.2 mM escitalopram oxalate, significantly increased apoptosis and elevated ROS levels were detected in SNU-1 cells. Additionally, a significantly reduced volume of the xenograft SNU-1 tumor was observed in nude mice. These findings proved the potentials of combinational use of low-dose 5-FU and escitalopram oxalate on inhibiting SNU-1 cells and may provide an alternative option for GC treatment.

## Figures and Tables

**Figure 1 ijms-23-16179-f001:**
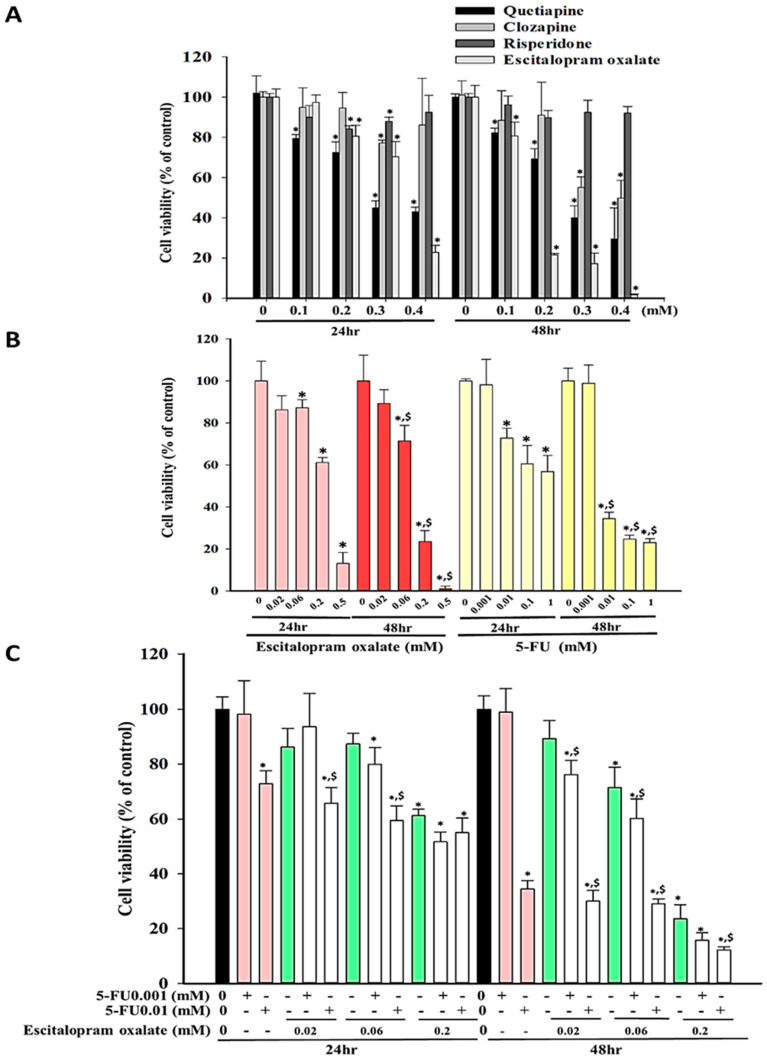
Effects of quetiapine, clozapine, risperidone, escitalopram oxalate, and 5–FU on the survival of SNU–1 cells. (**A**) The viability of SNU–1 cells in the presence of quetiapine, clozapine, risperidone, and escitalopram oxalate for 24 and 48 h. (**B**) The viability of SNU–1 cells in the presence of escitalopram oxalate and 5–FU for 24 and 48 h. (**C**) The viability of SNU–1 cells in the presence of the combinational use of escitalopram oxalate and 5–FU for 24 and 48 h. Statistical difference was calculated using one-way analysis of variance followed by Tukey’s multiple-comparisons test. The symbols * and $ indicate significant differences as compared with the control (0 mM) and escitalopram oxalate, respectively. The experiments were performed in triplicate. 5–FU: 5–fluouracil.

**Figure 2 ijms-23-16179-f002:**
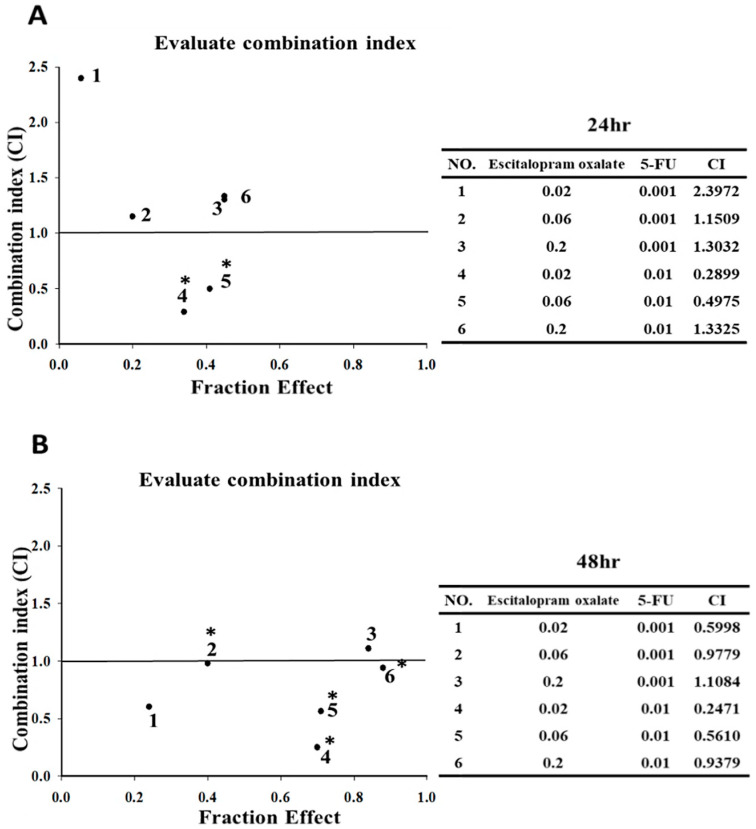
Evaluation of the CI. The CI values of 5–FU and escitalopram oxalate on the SNU–1 cells for (**A**) 24 and (**B**) 48 h. The symbol * indicates synergistic effects. CI: combination index; 5–FU: 5–fluouracil. The drug combinations 1–6 are shown in the tables.

**Figure 3 ijms-23-16179-f003:**
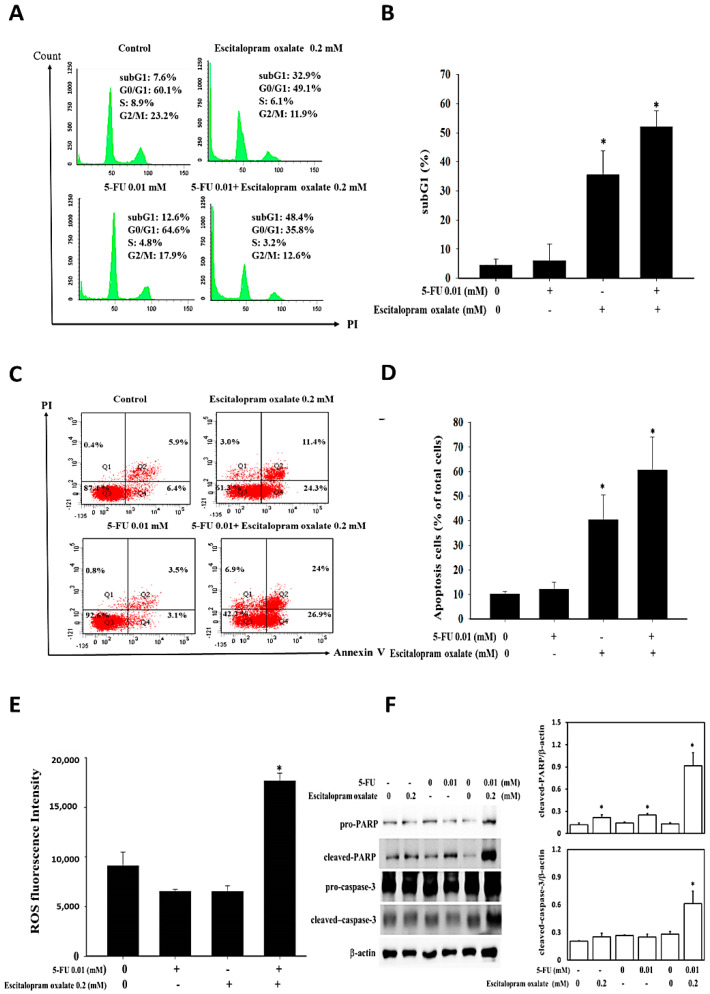
Effects of the combinational use of escitalopram oxalate and 5–FU on sub-G1 portion, apoptosis, ROS level and apoptotic proteins in SNU–1 cells. The SNU–1 cells were treated with escitalopram oxalate (0.2 mM) and 5–FU (0.01 mM) alone or in combination for 48 h. (**A**) Representative images of the cell cycle and (**B**) sub–G1 portion of the SNU–1 cells. (**C**) Representative images of annexin V/PI staining and (**D**) percentage of apoptotic cells. (**E**) Fluorescence intensity of ROS and (**F**) expressions of poly (ADP-ribose) polymerase and caspase–3 proteins. Right panel indicates the quantified results of Immunoblotting. Bars indicate mean ± SD from 3 repeated experiments. Statistical difference was calculated using one-way analysis of variance followed by Tukey’s multiple-comparisons test. The symbol * indicates significant differences as compared with the control (5–FU 0 mM and escitalopram oxalate 0 mM). 5–FU: 5–fluouracil.

**Figure 4 ijms-23-16179-f004:**
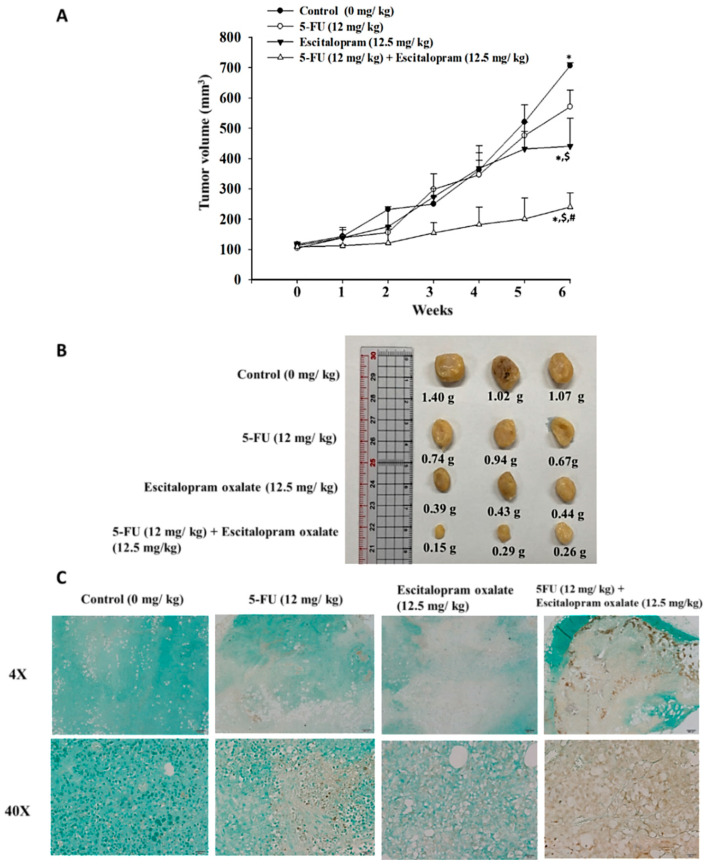
Effects of escitalopram oxalate + 5-FU on xenograft tumor growth in nude mice. (**A**) Volume of the xenograft SNU–1 tumor over a time-course pattern. (**B**) Representative images of the tumors excised from the mice at the experiment endpoints. (**C**) TUNEL staining of the xenograft tumors tissue sections obtained from different groups of mice. Statistical difference was calculated using one-way analysis of variance followed by Tukey’s multiple-comparisons test. The symbols *, $, and # indicate significant differences as compared with the control (0 mg/kg), 5–FU (12 mg/kg), and escitalopram oxalate (12.5 mg/kg), respectively. 5-FU: 5–fluouracil; TUNEL: Terminal deoxynucleotidyl transferase dUTP nick-end labeling.

## Data Availability

The data presented in this study are available on request from the corresponding author.

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
