# Peer review of "Synergistic Effects of the Combinational Use of Escitalopram Oxalate and 5-Fluorouracil on the Inhibition of Gastric Cancer SNU-1 Cells"

_ijms, 2022, doi:10.3390/ijms232416179_

Round 1

Reviewer 1 Report

Manuscript (ijms-2034203) provided by Chen et al. investigated the effect of combination of 5-FU in low concentrations with multiple concentrations of escitalopram oxalate on SNU-1 gastric cancer cell line. Moreover the author performed studies on nude mice xenograft model. All planned, evaluated and described studies performed by the authors of the manuscript were well planned and clearly described. The test results are clearly presented in graphs and figures. This research has great publishing potential and is of interest to IJMS readers. The authors proved the synergistic effect of the tested compounds on the studied gastric cancer cell line, which may constitute an interesting basis for further extended research. The reviewer does not see any major defects in the manuscript. The manuscript requires only small editorial corrections (periods, commas, spaces, superscripts and subscripts).

Reviewer 2 Report

In this manuscript, the authors tried to combine two drugs, the 5-florouracil and the escitalopram oxalate and showed a potential beneficial effect on cell proliferation and tumor growth. The concept is interesting but I have several comments for the authors.

Introduction:

I think that the introduction is too short. Some information on why it’s interesting to combine these two treatments, what studies have been already done, how the combination works, should be added.

Results:

Some figures should be merged together Fig. 1-2-3-4, Fig. 6-7-8. In my opinion is not relevant to make two panels for 24h and 48h for Fig. 1 to 5, these two time points should be merged in one graph.

Statistical tests must be described in figure legends

Transition and interpretation of some results are missing.

One cell line is not enough to conclude an interesting effect on gastric cancer

In my opinion XTT assay should not be used as the authors showed an effect of their treatments on ROS production.

·      Figure 2 

·      IC50 should be calculated using the clonogenic assay instead of XTT.

·      It’s atypical to see two IC50 for one compound

·      Figure 6

·      What the conclusion of this figure?

·      Figure 8

·      Quantification must be performed for the western blot

·      How the authors explained that there is not increase of cleaved PARP and Casp3 in escitalopram oxalate alone? The authors showed an increase of apoptosis previously for this condition 

Discussion:

I suggest to the authors to add a section of the possible interesting effect of escitalopram oxalate on 5-FU resistant cell lines.

Reviewer 3 Report

Repurposing used drugs for the cancer treatment is important part of medicinal reaserch. Topic of this manuscript is suitable for the IJMS and interesting for readers. Nevertheless, some poinst be taken for the improvement of manuscript.

Below works should be mentioned and discussed in the manuscript.

Enatescu VR, Papava I, Enatescu I, Antonescu M, Anghel A, Seclaman E, Sirbu IO, Marian C. Circulating Plasma Micro RNAs in Patients with Major Depressive Disorder Treated with Antidepressants: A Pilot Study. Psychiatry Investig. 2016 Sep;13(5):549-557. doi: 10.4306/pi.2016.13.5.549. Epub 2016 Sep 30. PMID: 27757134; PMCID: PMC5067350.

Hue JJ, Graor H, Zarei M, Katayama ES, Ji K, Hajihassani O, Loftus AW, Vaziri-Gohar A, Winter JM. IDO1 is a therapeutic target for pancreatic cancer-associated depression. Mol Cancer Ther. 2022 Oct 3:MCT-22-0055. doi: 10.1158/1535-7163.MCT-22-0055. Epub ahead of print. PMID: 36190971.

Chen LJ, Hsu TC, Chan HL, Lin CF, Huang JY, Stewart R, Tzang BS, Chen VC. Protective Effect of Escitalopram on Hepatocellular Carcinoma by Inducing Autophagy. Int J Mol Sci. 2022 Aug 17;23(16):9247. doi: 10.3390/ijms23169247. PMID: 36012510; PMCID: PMC9408912.

Shoaib M, Giacopuzzi E, Pain O, Fabbri C, Magri C, Minelli A, Lewis CM, Gennarelli M. Investigating an in silico approach for prioritizing antidepressant drug prescription based on drug-induced expression profiles and predicted gene expression. Pharmacogenomics J. 2021 Feb;21(1):85-93. doi: 10.1038/s41397-020-00186-5. Epub 2020 Sep 17. PMID: 32943772.

ielecka-Wajdman AM, Lesiak M, Ludyga T, Sieroń A, Obuchowicz E. Reversing glioma malignancy: a new look at the role of antidepressant drugs as adjuvant therapy for glioblastoma multiforme. Cancer Chemother Pharmacol. 2017 Jun;79(6):1249-1256. doi: 10.1007/s00280-017-3329-2. Epub 2017 May 12. PMID: 28500556.

IC50 values discussed in the 2.1 are different for shoved values in Fig. 2.

Minor

Page 2.1 cytotoxicity is also time dependently.

Figure 2. unit of IC50s are not shown

Page 12 “approximately 0.3 mM” in the plasma, serum?

Round 2

Reviewer 2 Report

1.     I’m still not convinced about the author’s choice for the layout of their figures.

2.     For the XTT, my point was that XTT assay measured cell viability by measuring extracellular reduction of XTT by NADH produced by mitochondria. The authors showed that their treatment combination increased ROS production. It’s well known that ROS can lead to mitochondria damage. Can the authors explained why they decided to use this method to measure cell viability instead of clonogenic assay? How to be sure that the decrease observed in XTT assay is due to effect of the treatment on cell viability and not an off-target of ROS production induced by the treatment?

3.     I disagree with the authors, XTT assay is not the most sensitive method as it indirectly measured cell viability by measuring mitochondrial activity. CellTox™ Green Cytotoxicity Assay or clonogenic assay are more sensitive.
